Positive Unlabeled Learning Selected Not At Random (PULSNAR): class proportion estimation without the selected completely at random assumption

http://orcid.org/0000-0002-4981-9020 Kumar Praveen
http://orcid.org/0000-0003-1994-2893 Lambert Christophe G. cglambert@salud.unm.edu
Department of Internal Medicine, Division of Translational Informatics, University of New Mexico , Albuquerque , United States
Moreo Alejandro
Electronic publication date: 2024 Nov 5
Publication date: 2024
Volume: 10
Electronic Location ID: e2451
Received 2024 May 15; Accepted 2024 Oct 4
Copyright: © 2024 Kumar and Lambert
Copyright year: 2024
Copyright holder: Kumar and Lambert
License: This is an open access article distributed under the terms of the Creative Commons Attribution License, which permits unrestricted use, distribution, reproduction and adaptation in any medium and for any purpose provided that it is properly attributed. For attribution, the original author(s), title, publication source (PeerJ Computer Science) and either DOI or URL of the article must be cited.
License URL: https://creativecommons.org/licenses/by/4.0/

Keywords: Positive-unlabeled learning, Noisy label learning, Class imbalance, Semi-supervised learning, PULSCAR, PULSNAR, Machine learning, SNAR, SCAR, Probability calibration, Class prior, Mixture proportion estimation

Funding: National Institute of Mental Health of the National Institutes of Health R01MH129764, R56MH120826 This research was supported by the National Institute of Mental Health of the National Institutes of Health under award numbers R01MH129764 and R56MH120826. The funders had no role in study design, data collection and analysis, decision to publish, or preparation of the manuscript.

==============================
Positive and unlabeled (PU) learning is a type of semi-supervised binary classification where the machine learning algorithm differentiates between a set of positive instances (labeled) and a set of both positive and negative instances (unlabeled). PU learning has broad applications in settings where confirmed negatives are unavailable or difficult to obtain, and there is value in discovering positives among the unlabeled (e.g., viable drugs among untested compounds). Most PU learning algorithms make the selected completely at random (SCAR) assumption, namely that positives are selected independently of their features. However, in many real-world applications, such as healthcare, positives are not SCAR (e.g., severe cases are more likely to be diagnosed), leading to a poor estimate of the proportion, α, of positives among unlabeled examples and poor model calibration, resulting in an uncertain decision threshold for selecting positives. PU learning algorithms vary; some estimate only the proportion, α, of positives in the unlabeled set, while others calculate the probability that each specific unlabeled instance is positive, and some can do both. We propose two PU learning algorithms to estimate α, calculate calibrated probabilities for PU instances, and improve classification metrics: i) PULSCAR (positive unlabeled learning selected completely at random), and ii) PULSNAR (positive unlabeled learning selected not at random). PULSNAR employs a divide-and-conquer approach to cluster SNAR positives into subtypes and estimates α for each subtype by applying PULSCAR to positives from each cluster and all unlabeled. In our experiments, PULSNAR outperformed state-of-the-art approaches on both synthetic and real-world benchmark datasets.

Introduction

1 In a standard binary supervised classification problem, the classifier (e.g., decision trees, support vector machines, etc.) is given training instances X with features x and their labels y=0 (negative) or y=1 (positive). The classifier learns a model f:X→0,1, which classifies an unlabeled instance as positive or negative based on x. It is often challenging, expensive, and even impossible to annotate large datasets in real-world applications (Jaskie, Elkan & Spanias, 2019), and frequently only positive instances are labeled. Unlabeled instances with their features can be classified via positive and unlabeled (PU) learning (Jaskie, Elkan & Spanias, 2019; Elkan & Noto, 2008). Some of the PU learning literature focuses on improving classification metrics, and others focus on the problem of estimating the fraction, α, of positives among the unlabeled instances. Although this work focuses on the latter, calibration and enhancing classification performance are also addressed.

PU learning problems abound in many domains (Jaskie & Spanias, 2019). For instance, in electronic healthcare records, the lack of a diagnosis code does not confirm a patient’s negative disease status, as negatives are not routinely recorded nor have billing codes, making traditional supervised learning impractical. Much medical literature is dedicated to estimating disease incidence and prevalence but contends with incomplete medical assessment and recording. The potential to assess disease incidence without costly in-person assessment or chart reviews could have substantial public health benefits. In market research, one typically has a medium-sized set of positives, say of customers or buyers of a product, has a set of attributes over both the positives and a large population of unlabeled people of size N, and wishes to establish the size of the addressable market, αN.

The majority of PU learning algorithms use the selected completely at random (SCAR) assumption, which states that the labeled positive examples are randomly selected from the universe of positives. That is, the labeling probability of any positive instance is constant (Elkan & Noto, 2008). This assumption may fail in real-world applications. For example, in email spam detection, positive instances labeled from an earlier time period could differ from later spam due to adaptive adversaries. In patient health data, for a given disease, labeled positive examples might be skewed to patients with severe symptoms who sought medical attention, while many mild cases remain undiagnosed and unlabeled. In fraud detection systems, labeled positive examples (fraudulent transactions) are those confirmed through investigations. Many fraudulent transactions go undetected and unlabeled. In social media bot detection, labeled bots are those reported by users or identified by automated systems. Many sophisticated bots go unnoticed and remain unlabeled.

Although some PU learning algorithms have shown promising performance on different machine learning (ML) benchmark SCAR datasets, the development of PU learning algorithms to estimate the extent of undercoding in large and highly imbalanced selected not at random (SNAR) real-world data remains an active research area. Class imbalance in a PU setting generally means the number of unlabeled instances is large compared to the labeled positive examples. Also, current PU learning approaches have rarely explored how to calculate well-calibrated probabilities for PU examples in SCAR and SNAR settings. In addition, few PU algorithms have been assessed when α is small (≤5%), where performance is expected to suffer.

In this article, we propose a PU learning approach to estimate α when positives are SCAR or SNAR, and evaluate its performance in simulated and real data. We assess the performance with class imbalance in both medium-sized and large datasets and over a rigorous α range. Our contributions are summarized as follows: 1) We propose PULSCAR, a PU learning algorithm for estimating α when the SCAR assumption holds. It uses kernel density estimates of the positive and unlabeled distributions of ML probabilities to estimate α. The algorithm employs the beta distribution to estimate density and introduces an objective function that provides a rapid, robust estimate of α.

2) We propose PULSNAR, a PU learning algorithm for estimating α when the positives are SNAR. It employs a clustering approach to group SNAR positives into subtypes and estimates separate α for each subtype using PULSCAR on positives from each cluster and all unlabeled instances. The final α is calculated by aggregating the α estimated for each subtype.

3) We propose methods to calibrate the probabilities of PU examples to their true (unknown) labels and improve the classification performance in SCAR and SNAR settings.

Related work

Early PU learning methods (Yu, Han & Chang, 2004; Yu, 2005; Wang et al., 2006) generally followed a two-step heuristic: (i) identify strong negative examples from the unlabeled set, and then (ii) apply an ML algorithm to given positive and identified negative examples. In contrast, Fung et al. (2005) extracted high-quality positive and negative examples from the unlabeled set and then applied classifiers to those data. Some recent work iteratively identifies better negatives (Luo et al., 2021), or combines negative-unlabeled learning with unlabeled-unlabeled learning (Hammoudeh & Lowd, 2020).

Studies predominantly centered around the SCAR assumption focused on estimating the proportion of positives among the unlabeled examples with or without the PU classifier. Elkan & Noto (2008) introduced the SCAR assumption and proposed a PU method to estimate the mixture proportion under the SCAR assumption. By partially matching the class-conditional density of the positive class to the input density under Pearson divergence minimization, Du Plessis & Sugiyama (2014) estimated the mixture coefficient. Jain et al. (2016) proposed a nonparametric class prior estimation technique, AlphaMax, using two-component mixture models. The kernel embedding approaches KM1 and KM2 (Ramaswamy, Scott & Tewari, 2016) showed that the algorithm for mixture proportion estimation converges to the true prior under certain assumptions. Estimating the class prior through decision tree induction (TIcE), Bekker & Davis (2018) provides a lower bound for label frequency under the SCAR assumption. Using the SCAR assumption, DEDPUL (Ivanov, 2020) estimates α by applying a compute-intensive EM-algorithm to probability densities; the method also returns uncalibrated probabilities.

Other studies focused on learning a classifier from PU data. Lee & Liu (2003) converts PU data learning into a noisy learning problem by designating all unlabeled instances as negatives. They employ a linear function to learn from these noisy examples using weighted logistic regression. Confident learning (CL) (Northcutt, Jiang & Chuang, 2021) combines the principle of pruning noisy data, probabilistic thresholds to estimate noise, and sample ranking. Multi-positive and unlabeled learning (Xu et al., 2017) extends PU learning to multi-class labels. Oversampling the minority class (Chawla et al., 2002; Yan et al., 2019) or undersampling the majority class are not well-suited approaches for PU data due to contamination in the unlabeled set; Su, Chen & Xu (2021) uses a re-weighting strategy for imbalanced PU learning.

Recent studies have focused on labeling/selection bias to address the SCAR assumption not holding. Bekker, Robberechts & Davis (2019), Gerych et al. (2022) used propensity scores to address labeling bias and improve classification. Using the propensity score, based on a subset of features, as the labeling probability for positive examples, Bekker, Robberechts & Davis (2019) reduced the Selected At Random (SAR) problem into the SCAR problem to learn a classification model in the PU setting. The “Labeling Bias Estimation” approach was proposed by Gong et al. (2021) to label the data by establishing the relationship among the feature variables, ground-truth labels, and labeling conditions.

Problem formulation and algorithms

In this section, we explain: i) the SCAR and SNAR assumptions, ii) our PULSCAR algorithm for SCAR data and PULSNAR algorithm for SNAR data, iii) kernel density and bandwidth estimation techniques, and iv) a method to find the number of clusters in the labeled positive set. Our method to calibrate probabilities and enhance classification performance using PULSCAR/PULSNAR is in “Algorithm for Calibrating Probabilities” and “Improving Classification Performance with PULSCAR and PULSNAR”, sections in the Appendix, respectively.

SCAR assumption and SNAR assumption

In PU learning settings, a positive or unlabeled example can be represented as a triplet ( x,y,s) where “ x” is a vector of the attributes, “ y” the actual class, and “ s” a binary variable representing whether or not the example is labeled. If an example is labeled ( s=1), it belongs to the positive class ( y=1) i.e., p(y=1|s=1)=1. If an example is not labeled ( s=0), it can belong to either class. Since only positive examples are labeled, p(s=1|x,y=0)=0 (Elkan & Noto, 2008). Under the SCAR assumption, a labeled positive is an independent and identically distributed (i.i.d.) example from the positive distribution, i.e., positives are selected independently of their attributes. Therefore, p(s=1|x,y=1)=p(s=1|y=1) (Elkan & Noto, 2008).

For a given dataset, p(s=1|y=1) is a constant and is the fraction of labeled positives. If |P| is the number of labeled positives, |U| is the number of unlabeled examples, and α is the unknown fraction of positives in the unlabeled set, then

(1) p(s=1)=|P||P|+|U|andp(y=1)=|P|+α|U||P|+|U|(classprior)p(s=1|y=1)=p(y=1|s=1)p(s=1)p(y=1)=p(s=1)p(y=1),sincep(y=1|s=1)=1=|P||P|+α|U|,whichisaconstant.

On the contrary, under the SNAR assumption, the probability that a positive example is labeled is not independent of its attributes. Stated formally, the assumption is that p(s=1|x,y=1)≠p(s=1|y=1) i.e. p(s=1|x,y=1) is not a constant.

Whether the SCAR assumption holds or fails can also be formalized using the conditional probability p(x|y=1). When the SCAR assumption holds, p(x|y=1,s=1)=p(x|y=1,s=0)=p(x|y=1), suggesting feature similarity between labeled and unlabeled positives in the data. The SCAR assumption holds if the positive class contains c subclasses, c1, c2, …, cc, in both labeled and unlabeled examples and for each subclass ci, p(x|y=1,ci,s=1)=p(x|y=1,ci,s=0). This condition ensures that the process of labeling does not introduce any bias within each subclass of the positive class. On the contrary, the SCAR assumption fails if for any subclass ci, p(x|y=1,ci,s=1)≠p(x|y=1,ci,s=0). This indicates that the selection process for labeled positives is not random within the subclass; specific features influence which positive examples get labeled, violating the SCAR assumption.

The SCAR assumption can hold when both labeled and unlabeled positives: (a) are not subclass mixtures, sharing similar attributes; (b) belong to c subclasses ( 1…c), with equal subclass proportions in both positive and unlabeled sets. Intra-subclass examples will have similar attributes, whereas the inter-subclass examples may not have similar attributes. E.g., in patients positive for diabetes, type 1 patients will be in one subclass, and type 2 patients will be in another. The SCAR assumption can fail when labeled and unlabeled positives are from c subclasses, and the proportion of those subclasses is different in positive and unlabeled sets. Suppose both positive and unlabeled sets have subclass 1 and subclass 2 positives, and in the positive set, their ratio is 30:70. If the ratio is also 30:70 in the unlabeled set, the SCAR assumption will hold. If it was different, say 80:20, the SCAR assumption would not hold.

PU data assumptions

Our proposed PU algorithms rely on the following assumptions about the PU data: 1) The labeled set consists exclusively of correctly labeled positive examples, while the unlabeled set contains a mixture of both positive and negative examples. That is, p(s=1|x,y=0)=0 (Elkan & Noto, 2008). 2) Unlabeled positive examples have support (examples with similar features) in the labeled positive set, i.e., p(x|y=1,s=1)=p(x|y=1,s=0). In the SNAR case where there are multiple subclasses of positives, we further assume that each subclass among the unlabeled set must have support in the labeled set. Other formulations of the SNAR case may be conceivable, but this assumption applies to the SNAR formulation we have considered in this manuscript. If the second assumption does not hold, our approaches may estimate an inaccurate α, often resulting in underestimates. It is important to clarify that these assumptions are not novel propositions introduced by our algorithms. Assumption 1 is a basic aspect of PU learning, while Assumption 2 follows the definition of SCAR.

In the first step of our PU algorithms, we execute a supervised probabilistic classifier (e.g., XGBoost; Chen & Guestrin, 2016) with five-fold cross-validation (CV) by treating all labeled positives as class 1 and all unlabeled examples as class 0. The goal is to obtain classifier-estimated class 1 probabilities for all PU examples to determine α, the fraction of positives among the unlabeled examples. It is worth noting that in our PU algorithms, α is not the same as the class prior. The class prior represents the proportion of positives, labeled and unlabeled combined, in the dataset and can be estimated if α is known.

Positive and Unlabeled Learning Selected Completely At Random (PULSCAR) algorithm

Given any supervised probabilistic classifier, A(x,y), let Dp, Dn, and Du be probability density functions (PDFs) corresponding to the probability distribution of positives, negatives, and unlabeled respectively. Let α be the unknown proportion of positives in the unlabeled, then

(2) Du=αDp+(1−α)Dn⇒Du≥αDp,since0≤α≤1andDnisnon-negative⇒0≤αDp≤Du,sinceDpisnon-negative

From relation (Eq. (2)), a key observation is that αDp should not exceed Du, ensuring that the α is bounded. However, the true α is usually difficult to identify (Ivanov, 2020; Blanchard, Lee & Scott, 2010). Therefore, PULSCAR aims to estimate the largest α value ( αmax) that ensures Dui−αDpi remains non-negative throughout the range, i ∈ [0 …1]. We calculate Du and Dp for n_bins equidistant points within the range [0, 1]. Thus, Du and Dp are vectors that represent discrete approximations of the continuous PDFs. The variable i represents each of these n_bins points. The αmax is given by:

αmax=min(DuiDpi)foralli,whereDpi≠0

Leveraging relation (Eq. (2)), we define the following objective function (Eq. (3)) that PULSCAR utilizes to estimate α:

(3) f(α)=log(|min(Du−αDp)|+ϵ),whereϵ=|min(Dp)|ifmin(Dp)≠0, else ϵ=10−10

Explanation of function components: Since Du and Dp are vectors, we identify the smallest difference between Du and αDp for each α value in [0 …1] using the min() function. This ensures the objective function is not significantly affected by potential outliers in Du or Dp. The absolute value ensures we consider the magnitude of this minimum difference, regardless of whether Du is greater or less than αDp. It also prevents taking the logarithm of a negative value. The log() function makes the objective function more sensitive to small differences between Du and αDp, and the inclusion of ϵ prevents log(0) when |min(Du−αDp)| approaches zero.

How PULSCAR estimates α: We train a supervised probabilistic classifier (e.g., XGBoost) treating all labeled positives as class 1 and all unlabeled examples as class 0 to estimate the class 1 probabilities for all instances. We then calculate Dp and Du using the beta kernel density method on estimated class 1 probabilities for positive and unlabeled examples, respectively. The intuition behind the objective function is that |min(Du−αDp)| approaches zero at an α point where αDp equals Du, causing the objective function to tend toward −∞, see Fig. 1D. Consequently, when |min(Du−αDp)| is not zero, there is a maximum change in the value of the objective function (from −∞ to some value). Thus, this α value creates a sharp change in the objective function, which usually coincides with αmax because, at αmax, Du is closest to αDp. To estimate α, PULSCAR searches for the point of maximum change in the objective function using its slope. We calculate the value of the objective function, f(αi), for each αi in [0 …1] with a step size of 0.0001. We then apply the finite difference method to approximate the slope of the objective function at each α value in this range. The α value corresponding to the maximum absolute difference in slope is selected as the estimated α. The objective function may not be convex for some datasets, i.e., there could be multiple points of sharp change in the objective function. In such cases, we select the α corresponding to the first point of sharp change (closest to zero) because the term Du−αDp, representing the estimated PDF of negative examples, must be non-negative. Selecting a point of sharp change in the objective function beyond the first one may yield a larger α that can make Du−αDp negative, violating PDF constraints (Fig. 1B). This approach eliminates the need for implementing an iterative solver technique, accounting in part for the speed of our algorithm. The value of α can also be determined visually by plotting the objective function (Fig. 1D); the sharp change in the plot represents the value of α. Algorithm 1 shows the pseudocode of the PULSCAR algorithm to estimate α using the objective function based on probability densities. Algorithm 2 is a subroutine to compute the beta kernel bandwidth. Full source code for our algorithms is available at GitHub (https://github.com/unmtransinfo/PULSNAR/).

Figure 1 PULSCAR algorithm visual intuition.

PULSCAR finds the largest α such that the estimated density of negative examples, Du−αDp, never falls below zero in [0…1]. (A) Probability density estimates for simulated data with α=10% positive examples in the unlabeled set–estimated density of negative examples (blue) nearly equals the ground truth (green). (B) Overweighting the density of positive examples by α=15% results in the estimated density of negative examples (blue), dropping below zero. (C) Underweighting the density of positive examples by α=5% results in the estimated density of negative examples (blue) being higher than the ground truth (green). (D) Estimated α=10.68% selected where the absolute difference in slope of the objective function is largest–very close to ground truth α=10%. Du: PDF of all unlabeled examples (black, dotted); αDp: PDF of unlabeled positive examples scaled by α (orange, dotdash); Du−αDp: estimated PDF of unlabeled negative examples (blue, solid); (1−α)Dn: true PDF of unlabeled negative examples scaled by (1−α) (green, twodash).

Algorithm 1 PULSCAR algorithm.

Input: X = Xp∪Xu, s = sp∪su, n_bins	
{‘s’ indicates assigned labels (labeled positives: s=1, unlabeled: s=0); n_bins is number of bins}	
Output: estimated α	
  1: p ←A(X, s) { A(X, s) returns estimated class 1 probabilities for all PU examples}	
  2: pu← p[s==0] {s==0 indicates unlabeled indices}	
  3: pp← p[s==1] {s==1 indicates labeled positive indices}	
  4: estimation_range ← [0, 0.0001, 0.0002,…, 1.0]	
  5: bw ← estimate_bandwidth_pu(p, n_bins)	
  6: Du← beta_kernel(pu, bw, n_bins)	
  7: Dp← beta_kernel(pp, bw, n_bins)	
  8: ε←|min(Dp)|	
  9: if ε = 0 then	
 10:    ϵ←10−10	
 11: end if	
 12: len ← length(estimation_range)	
 13: selected_range ← estimation_range[2:len]	
 14: α← estimation_range	
 15: f(α) ←log(|min(Du−αDp)|+ϵ)	
 16: d ← f’(α)	
 17: i ← where the absolute difference in d is the maximum	
 18: return selected_range[i]	

Algorithm 2 Estimate bandwidth (estimate_bandwidth_pu).

Input: preds, n_bins	
{‘preds’ indicates the classifier-estimated class 1 probabilities}	
Output: bandwidth	
 1: bw ∈ [0.01, 0.5]	
 2: Dhist← histogram(preds, n_bins, density=True)	
 3: Dbeta← beta_kernel(preds, bw, n_bins)	
 4: return optimize(MeanSquaredError(Dhist, Dbeta))	

Our PULSCAR algorithm (Algorithm 1) uses a histogram bin count heuristic to generate a histogram-derived density, then optimizes the beta distribution kernel bandwidth to best fit that density. The approach relies on three key elements: the use of the beta kernel for density estimates, the histogram bin count, and the kernel bandwidth. The beta kernel has important properties, described below, for fitting distributions over the interval [0…1]. The histogram bin count parameter is set using standard heuristics for fitting a histogram to data, and defines an array of evenly spaced numbers over the interval [0…1], where the beta kernel density estimates are computed to optimize fitting the histogram. The bandwidth parameter determines the width of the beta kernel, balancing the need for narrower kernels to precisely estimate the distribution with many data points against wider kernels to avoid overfitting when data are sparse. The following three subsections detail these parameters and explain how they are determined.

Beta kernel density estimation

A beta kernel estimator is used to create a smooth density estimate of both the positive and unlabeled ML probabilities, generating distributions over [ 0…1], free of the problematic boundary biases of kernels (e.g., Gaussian) whose range extends outside that interval, adopting the approach of Chen (1999). Another problem with (faster) Gaussian kernel density implementations is that they often use polynomial approximations that can generate negative values in regions of low support, dramatically distorting α estimates, which require non-negative probability distribution estimates. The beta PDF is as follows (Virtanen et al., 2020):

(4) h(x,a,b)=Γ(a+b)xa−1(1−x)b−1Γ(a)Γ(b),

for x ∈ [0,1], where Γ is the gamma function, a=1+zbw and b=1+1−zbw, with z the bin edge (a value in an array of evenly spaced n_bins numbers over the interval [0, 1]), and bw the bandwidth.

Histogram bin count

Our implementation supports five well-known methods to determine the number of histogram bins: square root, Sturges’ rule, Rice’s rule, Scott’s rule, and Freedman–Diaconis (FD) rule (Alxneit, 2020). Let ‘pr’ be the classifier-estimated class 1 probabilities for all PU examples and ‘n’ be the number of PU examples. The formulae to count the number of bins using these methods are as follows: i) Square root method: Numberofbins=n

ii) Sturges’ rule: Numberofbins=1+log2(n)

iii) Rice’s rule: Numberofbins=2×n1/3

iv) Scott’s rule:

h=3.5×StandardDeviation(pr)n1/3, Numberofbins=max(pr)−min(pr)h

v) Freedman–Diaconis (FD) rule:

h=2×InterQuartileRange(pr)n1/3, Numberofbins=max(pr)−min(pr)h

Beta kernel bandwidth estimation

The bandwidth of the kernel is the smoothing parameter in kernel density estimation that affects the shape of the estimated distribution curve. A broader bandwidth results in a smoother and more generalized curve, whereas a narrower bandwidth produces a more fine-grained curve. Kernel bandwidth selection can also significantly influence α estimates: too narrow of a bandwidth can result in outliers driving poor estimates, and too wide of a bandwidth prevents distinguishing between distributions. We compute a histogram-derived density using the aforementioned bin count heuristic and beta kernel density estimate at those bin centers using the ML probabilities of both the positive and unlabeled examples. We find the minimum of the mean squared error (MSE) between the histogram and beta kernel densities using the scipy differential_evolution() optimizer (Storn & Price, 1997), solving for the best bandwidth in the range [0.01…0.5]. The differential_evolution() algorithm is a stochastic direct search method to find the minimum of a multivariate function without requiring gradient information. The estimated bandwidth is chosen for kernel density estimation in the PULSCAR algorithm. All experiments herein use MSE as the error metric, but alternatively, the Jensen-Shannon distance can be employed.

Positive and Unlabeled Learning Selected Not At Random (PULSNAR) algorithm

We propose a new PU learning algorithm (PULSNAR) to estimate the α in SNAR data, i.e., labeled positives are not selected completely at random. PULSNAR uses a divide-and-conquer strategy for the SNAR data to convert a SNAR problem into several sub-problems. It employs clustering techniques to divide SNAR positives into clusters, where each cluster predominantly contains one subtype of positives. Subsequently, it applies the PULSCAR algorithm to the positives from each cluster along with all unlabeled data to estimate the α for each subtype. The final α is computed by summing the α returned by the PULSCAR algorithm for each subtype of positives.

(5) α=α1+α2+...+αc,c=numberofclusters

Figure 2 visualizes the PULSNAR algorithm, and Algorithm 3 provides its pseudocode.

Figure 2 Schematic of PULSNAR algorithm.

An ML model is trained and tested with five-fold cross-validation (CV) on all positive and unlabeled examples. The important covariates that the model used are scaled by their importance value. Positives are divided into c clusters using the scaled important covariates. c ML models are trained and tested with five-fold CV on the records from a cluster and all unlabeled records. We estimate the proportions ( α1...αc) of each subtype of positives in the unlabeled examples using PULSCAR. The sum of those estimates gives the overall fraction of positives in the unlabeled set. P = positive set, U = Unlabeled set.

Algorithm 3 PULSNAR algorithm.

Input: X = Xp∪Xu, s = sp∪su, n_bins	
{‘s’ indicates assigned labels (labeled positives: s=1, unlabeled: s=0); n_bins is number of bins}	
Output: estimated α	
 1: feature_importance ( v1…vk), imp_features ( x1…xk) ←A(X, s) { A(X, s) returns important features and their importance (gain scores)}	
 2: x1′…xk′←x1v1…xkvk {scale important features by their gain score}	
 3: Xp′←Xp[ x1′…xk′] {select labeled positives with only scaled important features}	
 4: clusters C1...Cc← GMM( Xp′) {divide labeled positives into c clusters}	
 5: α← 0	
 6: for j in C1...Cc do	
 7:    X′←Xp[j] ∪Xu {select labeled positives from cluster and all unlabeled}	
 8:    s′←sp[j] ∪su	
 9:    α←α+PULSCAR(X′,s′,n_bins)	
10: end for	
11: return α	

Clustering rationale

Suppose both the positive and unlabeled sets contain positives from c subclasses ( 1…c). With selection bias (SNAR), the subclass proportions will vary between the sets, and thus, the PDF of the labeled positives cannot be scaled by a uniform α to estimate positives in the unlabeled set. The smallest subclass would drive an α underestimate with PULSCAR. To address this, we perform clustering on the labeled positives using the key features that differentiate positives from the unlabeled ones. This assumes that these features also correlate with the factors driving selection bias within each subclass. By dividing positives into c clusters based on these features, we establish that within each cluster, the selection bias is uniformly distributed, implying that all members of a cluster are equally likely to appear in the unlabeled set. Consequently, PU data comprising examples from one cluster and the unlabeled set approximates the SCAR assumption. The uniformity allows for the inference of a cluster-specific α using PULSCAR, which can be summed over all clusters to provide an overall estimate of positives in the unlabeled set (Fig. 2).

Determining the number of clusters in the positive set

Determining the “optimal number of clusters” is a challenging problem, as evidenced by the NP-hardness of optimal k-means clustering in the literature (Mahajan, Nimbhorkar & Varadarajan, 2012). Our approach to determining the number of clusters via BIC is one of the widely used heuristics and approximation methods (Zhao, Hautamaki & Fränti, 2008; Ezugwu et al., 2022; Fraley & Raftery, 1998). We build an XGBoost model on all positive and unlabeled examples to determine the important features and their gain scores. A gain score measures the magnitude of the feature’s contribution to the model. We select all labeled positives and then cluster them on those features scaled by their corresponding gain score, using scikit_learn’s Gaussian mixture model (GMM) method. To establish the number of clusters (n_components), we iterate n_components over 1…c (e.g., c = 25) and compute the Bayesian information criterion (BIC) (Vrieze, 2012) for each clustering model. We use max_iter = 250, and covariance_type = “full”. The other parameters are used with their default values. We implemented the “Knee Point Detection in BIC” algorithm, explained in Zhao, Hautamaki & Fränti (2008), to find the number of clusters in the labeled positives.

Calculating calibrated probabilities

The approach to calibrate the classifier-estimated probabilities of positive and unlabeled examples in the SCAR and SNAR data is explained in “Algorithm for Calibrating Probabilities” section of the Appendix.

Improving classification performance

Enhancing classification with PULSCAR/PULSNAR involves estimating α and using the resulting calibrated probabilities to re-label the top α|U| unlabeled examples as positives, while treating the remainder of the unlabeled examples as negatives. The original positives retain their labels. Standard ML classification techniques (e.g., XGBoost) are then employed on these updated labels, as detailed in “Improving Classification Performance with PULSCAR and PULSNAR” section of the Appendix.

Experimental methods

We evaluated our proposed PU learning algorithms in terms of α estimates, probability calibration (“Experiments and Results”), and six classification performance metrics (“Experiments and Results”). We used real-world ML benchmark datasets and synthetic data for our experiments. For real-world data, we used KDD Cup 2004 particle physics (Caruana, Joachims & Backstrom, 2004), Diabetes health indicators (UCI ML Repository, 2024), Magic gamma telescope (Bock, 2007), Electrical grid stability (Arzamasov, 2018), Wilt (Johnson, 2014), and Mice protein expression (Higuera, Gardiner & Cios, 2015) as SCAR datasets and Anuran calls (Colonna et al., 2017), dry bean (UCI Machine Learning Repository, 2020), Room occupancy estimation (Singh & Chaudhari, 2023), Smartphone (Davis & Owusu, 2016), Letter recognition (Slate, 1991), and Statlog (Shuttle) (UCI Machine Learning Repository, 2022) as SNAR datasets. Synthetic (SCAR and SNAR) datasets were generated using the scikit-learn function make_classification() (Pedregosa et al., 2011). We used XGBoost as a binary classifier in our proposed algorithms. We evaluated PU algorithms on imbalanced SCAR and SNAR datasets. In some datasets, we kept the number of unlabeled examples greater than the positive examples while in others, we kept it lesser. By setting the number of unlabeled examples to be less than positive examples, we were able to test the PU algorithms for a higher proportion of positives among unlabeled examples in ML benchmark datasets. To address class imbalance, we used the scale_pos_weight parameter of XGBoost to scale the weight of the labeled positive examples by the factor s=|U||P|. We also compared our methods with recently published methods for PU learning: KM1 and KM2 (Ramaswamy, Scott & Tewari, 2016), TIcE (Bekker & Davis, 2018) and DEDPUL (Ivanov, 2020). Some recent studies (Bekker, Robberechts & Davis, 2019; Gerych et al., 2022) investigating scenarios where SCAR does not hold do not focus on α estimation or probability calibration. Therefore, we opted to exclude them from our comparison. KM1, KM2, and TIcE algorithms were not scalable on large datasets, so we used relatively smaller synthetic and ML benchmark datasets to compare our methods with these methods. We compared PULSNAR with only DEDPUL on large synthetic datasets (“DEDPUL vs. PULSNAR: Alpha Estimation”). Also, Ivanov (2020) previously demonstrated that DEDPUL outperformed KM and TIcE algorithms on several ML benchmark and synthetic datasets.

Synthetic data

We generated SCAR and SNAR PU datasets with different fractions of positives (1%, 5%, 10%, 20%, 30%, 40%, and 50%) among the unlabeled examples to test the effectiveness of our proposed algorithms. For each fraction, we generated 40 datasets using sklearn’s make_classification() function with random seeds 0–39. The class_sep parameter of the function was used to specify the separability of data classes. Values nearer to 1.0 make the classification task easier; we used class_sep = 0.3 to create difficult classification problems.

SCAR data

The datasets contained 2,000 positive (class 1) and 6,000 unlabeled (class 0) examples with 50 continuous features. The unlabeled set comprised k% positive examples with labels flipped to 0 and (100−k)% negative examples.

SNAR data

We generated datasets with six labels (0–5), defining ‘0’ as negative and 1–5 as positive subclasses. These datasets contained 2,000 positives (400 from each positive subclass) and 6,000 unlabeled examples with 50 continuous features. The unlabeled set comprised k% positive examples with labels (1–5) flipped to 0 and (100 − k)% negative examples. The unlabeled positives were markedly SNAR, with the five subclasses comprising 1/31, 2/31, 4/31, 8/31, and 16/31 of the unlabeled positives (e.g., in the unlabeled set with 20% positives, negative: 4,800, label 1 positive: 39, label 2 positive: 77, label 3 positive: 155, label 4 positive: 310, label 5 positive: 619).

ML benchmark datasets

Table 1 presents a comprehensive overview of the benchmark datasets used for testing PU methods. To add k% positive examples to the unlabeled set, the labels of m randomly selected positive records were flipped from 1 to 0, where m=k|U|100−k. In order to satisfy the SNAR assumption, positive examples were selected from every positive subclass for label flipping while ensuring a distinct ratio of positive examples in the positive and unlabeled sets in SNAR datasets.

Table 1 List of SCAR and SNAR datasets with number of records and features.

The table also shows which class was used as negative and which class(es) was/were used as positive. Unlabeled records included all records from the negative class and fractions of records from the positive class(es).

SCAR datasets	Record count	Feature count	Positive class	Negative class	
KDD cup 2004 particle physics	50,000	77	1	0	
CDC diabetes health indicators	253,680	21	0	1	
Magic gamma telescope	19,020	10	g	h	
Electrical grid stability simulated data	10,000	13	Unstable	Stable	
Wilt	4,839	5	n	w	
Mice protein expression	1,080	77	c-CS-m	t-SC-s	
SNAR datasets	
Anuran calls	7,127	23	Leptodactylidae, Dendrobatidae	Hylidae	
Dry bean	13,611	16	SIRA, SEKER, HOROZ, CALI, BARBUNYA, BOMBAY	DERMASON	
Room occupancy estimation	10,129	16	0, 1, 3	2	
Smartphone dataset for human activity recognition	7,415	561	1, 2, 3, 4, 6	5	
Letter recognition	4,639	16	A, B, C, E, F	D	
Statlog (Shuttle)	43,500	9	2, 3, 4, 5, 6, 7	1	

Estimation of fraction of positives among unlabeled examples

We applied the PULSCAR algorithm to both SCAR and SNAR data, and the PULSNAR algorithm only to SNAR data, to estimate α.

Using the PULSCAR algorithm

To find the 95% confidence interval (CI) on estimation, we ran XGBoost with five-fold CV for 40 random instances of each dataset generated (or selected from benchmark data) using 40 random seeds. Each iteration’s class 1 estimated probabilities of positives and unlabeled were used to calculate the value of α.

Using the PULSNAR algorithm

The labeled positives were divided into c clusters to obtain homogeneous subclasses of labeled positives. The XGBoost ML models were trained and tested with five-fold CV on data from each cluster and all unlabeled records. For each cluster, α was estimated by applying the PULSCAR method to class 1 estimated probabilities of positives from the cluster and all unlabeled examples. The overall proportion was calculated by summing the estimated α for each cluster. To compute the 95% CI on the estimation, PULSNAR was repeated 40 times on data generated/selected using 40 random seeds.

Results

Synthetic datasets

Figure 3 shows the α estimated by PU learning algorithms for synthetic datasets. TIcE overestimated α for all fractions in both SCAR and SNAR datasets. For SCAR datasets, only PULSCAR returned close estimates for all fractions; DEDPUL overestimated for 1%; KM1 overestimated for ≥20%; KM2 underestimated for 50% and overestimated for 1%. For SNAR datasets, only PULSNAR’s estimates were close to the true α; other algorithms either overestimated or underestimated α for some or all fractions. Given the five subclasses in the simulated SNAR data, PULSNAR always estimated either five or six clusters, resulting in accurate overall α estimates when the subclass α estimates were summed. “DEDPUL vs. PULSNAR: Alpha Estimation” shows the α estimated by DEDPUL and PULSNAR on large SNAR datasets with different class imbalances. As the class imbalance increased, DEDPUL underestimated α, especially for larger fractions. The estimated α by the PULSNAR method was close to the true fractions across all sample sizes and class imbalances.

Figure 3 KM1, KM2, TIcE, DEDPUL, PULSCAR, and PULSNAR evaluated on SCAR and SNAR synthetic datasets.

The bar represents the mean value of the estimated α, with 95% confidence intervals for estimated α. The best estimators are close to the black bars, representing the true α. Bars larger than the black bars represent overestimation, while bars smaller than the black bars represent underestimation.

ML benchmark datasets

SCAR data

Figure 4 shows the α estimated by PU learning algorithms for SCAR benchmark datasets. The KDD cup and diabetes datasets were too large to execute KM1 and KM2. These methods produced overestimated α values for the magic gamma and electric grid datasets, and for the wilt and mouse protein datasets, α values were either overestimated or underestimated for some fractions. TIcE overestimated α for the KDD cup and mouse protein datasets, while for other datasets, it produced either overestimated or underestimated α values for some fractions. PULSCAR and DEDPUL both provided estimates that were close to the true answers for all fractions and datasets, but DEDPUL overestimated α for 1% for all datasets.

Figure 4 (A-F) KM1, KM2, TIcE, DEDPUL, and PULSCAR evaluated on SCAR ML benchmark datasets.

The bar represents the mean value of the estimated α, with 95% confidence intervals for estimated α. KM1 and KM2 failed to execute on the KDD cup and Diabetes datasets. The best estimators are close to the black bars, representing the true α. Bars larger than the black bars represent overestimation, while bars smaller than the black bars represent underestimation.

SNAR data

Figure 5 shows the α estimated by PU learning algorithms for SNAR benchmark datasets. For all datasets, only PULSNAR provided close estimates for all fractions, while other algorithms (KM1, KM2, TIcE, DEDPUL, and PULSCAR) either overestimated or underestimated α for some or all fractions. For fractions ≥15%, KM1, KM2, TIcE, DEDPUL, and PULSCAR usually underestimated α across all datasets. The α estimates of KM1 were inaccurate for all fractions except 1%, across all datasets. KM2 and TIcE showed inconsistent results, overestimating or underestimating even for smaller fractions ( <10%) in some datasets. DEDPUL overestimated alpha for 1% in all datasets. By contrast, our PULSCAR method provided α estimates that were close to the true answers for smaller fractions ( <10%).

Figure 5 (A-F) KM1, KM2, TIcE, DEDPUL, PULSCAR, and PULSNAR evaluated on SNAR ML benchmark datasets.

The bar represents the mean value of the estimated α, with 95% confidence intervals for estimated α. As KM1 and KM2 were taking several hours to finish one iteration on the Shuttle dataset, the mean α was computed using five iterations, and the standard error was set to 0. The best estimators are close to the black bars, representing the true α. Bars larger than the black bars represent overestimation, while bars smaller than the black bars represent underestimation.

Probability calibration

“Experiments and Results” contrasts the calibration curves for uncalibrated (blue) vs. isotonically calibrated (red) probabilities. The analysis further distinguishes whether the calibration included both original positives and unabeled examples, or just the unlabeled examples. Calibration for SCAR data utilized PULSCAR, while SNAR data calibration used PULSNAR. Calibration curves more closely aligned with the ideal y=x line for SCAR data as compared to SNAR data. Moreover, calibration results were superior for larger fractions as opposed to smaller fractions (1%) for both SCAR and SNAR data.

Classification performance metrics

“Experiments and Results” shows substantial improvement in six classification performance metrics when applying PULSCAR and PULSNAR vs. XGBoost alone. “DEDPUL vs. PULSNAR: Classification Performance” presents the classification performance of PULSNAR and DEDPUL on two real-world SNAR datasets. The classification performance of DEDPUL declined for larger fractions due to the underestimation of α. Conversely, PULSNAR exhibited robust classification performance due to its precise estimation of α, which were close to true fractions.

Execution time of PU methods

Table 2 presents the execution times of PU methods applied to synthetic SNAR and diabetes SCAR data. Despite PULSNAR determining the number of clusters and estimating α for each cluster, both PULSCAR and PULSNAR executed faster than other algorithms on large datasets. The specifications of the machine used to measure the execution time of these algorithms are as follows: Dual CPU AMD EPYC 9654 and 1.5TB RAM.

Table 2 Execution time (in minutes) of PU algorithms on synthetic and benchmark datasets based on a single iteration to estimate α for all true fractions we tested.

For synthetic SNAR data, we tested these methods on true fractions (0.01, 0.05, 0.10, 0.20, 0.30, 0.40, 0.50), while for diabetes SCAR data, we tested these methods on true fractions (0.01, 0.05, 0.10, 0.20, 0.30, 0.40, 0.50, 0.60, 0.70, 0.80). DEDPUL, PULSCAR, and PULSNAR were executed with 16 cores, whereas TIcE and KM do not provide an option to set the number of CPU cores and use all available cores. The publicly available source code for KM algorithms is not equipped to manage large datasets, resulting in program crashes.

Datasets	KM	TIcE	DEDPUL	PULSCAR	PULSNAR	
Synthetic SNAR data (P: 5,000, U: 100,000)	Failed	706.50	159.59	3.19	14.07	
Synthetic SNAR data (P: 5,000, U: 50,000)	Failed	349.28	42.04	1.77	7.91	
Synthetic SNAR data (P: 5,000, U: 10,000)	243.47	83.78	2.29	0.61	2.04	
Diabetes SCAR data (P: 218,334, U: 35,346)	Failed	1,100.73	199.88	18.31	27.85	

Discussion and conclusion

This article presented novel PU learning algorithms for estimating the proportion of positives ( α) among unlabeled examples in both SCAR and SNAR data with and without class imbalance. By utilizing the law of total probability to define the objective function (Eq. (3)) and a beta kernel to estimate probability densities, the PULSCAR algorithm estimates an upper bound on the fraction of positives in the unlabeled set under the assumption of SCAR and that the classes are separable. Although theoretically an upper bound, in practice, our estimates were all close to the actual ground truth α, albeit with greater percent overestimation with smaller α values. We observed in nearly all of our simulations on SCAR data that the α estimates were above ground truth, suggesting the PULSCAR algorithm is trustworthy for bounding α from above. Overestimation may occur due to errors in estimating the continuous density functions with finite sample sizes, and issues of lack of support between positives and unlabeled in the data generation process due to random sampling.

In our experiments, we demonstrated that our PULSCAR method outperformed other state-of-the-art methods, such as KM1, KM2, and TIcE, for estimating α on both synthetic and real-world SCAR datasets (Figs. 3A, 4). The rather poor performance of KM1, KM2, and TIcE in some datasets seems to be linked to the difficulty of the classification problems. Our simulated data used a low class separability (class_sep = 0.3), and experiments we did with higher class separability (class_sep = 0.9) resulted in better α estimation for those algorithms. To highlight the effectiveness of our algorithms under challenging conditions, we focused on scenarios with low class separability (class_sep = 0.3). We ran experiments with higher separability (class_sep = 0.9) in only a few simulated cases before discontinuing that line of inquiry. Consequently, results based on high separability were excluded from this article. The performance of the PULSCAR and DEDPUL methods was comparable on SCAR datasets, given their similar theoretical foundations, as well as the use of powerful classifiers (XGBoost for PULSCAR, CatBoost for DEDPUL). However, we observed that DEDPUL tends to have large overestimates for smaller fractions (e.g., for 1%, DEDPUL estimated an α of ≥2% for all datasets). The version of PULSCAR presented in this article uses the beta distribution, outperforming earlier prototypes that used the Gaussian distribution. This may account for PULSCAR’s superiority over DEDPUL for 1%, which uses the Gaussian kernel for density estimation.

Our PULSNAR method demonstrated superior performance over all other methods (KM1, KM2, TIcE, and DEDPUL) for estimating α across both synthetic and real-world SNAR datasets (Figs. 3B, 5, “DEDPUL vs. PULSNAR: Alpha Estimation”). Additionally, the PULSNAR method showed better classification performance compared to DEDPUL when evaluated on SNAR datasets (“DEDPUL vs. PULSNAR: Classification Performance”), particularly for larger fractions ( ≥20%). Interestingly, as the number of positive subclasses increased, the SCAR-based methods produced poor α estimates on SNAR data (Figs. 3B, 5), indicating that PU learning methods based on the SCAR assumption are not suitable for SNAR data.

Probability calibration plays a crucial role in several classification tasks, such as medical diagnosis and financial risk assessment. It helps determine the optimal probability threshold for decision-making, which ultimately enhances the reliability and usefulness of ML models (Silva Filho et al., 2023; Pleiss et al., 2017). For instance, in a binary classification scenario where model estimations determine whether a patient has a disease, the selection of an optimal probability threshold for making informed decisions relies on having calibrated probabilities. Our study presented a novel approach to calculating calibrated probabilities for positive and unlabeled examples, which, to the best of our knowledge, has not been explored in existing PU methods. Our experimental results demonstrated that both PULSCAR and PULSNAR methods significantly improved probability calibration for SCAR and SNAR datasets, respectively (“Experiments and Results”).

While not the main focus of this work, we also demonstrated that after applying PULSCAR and PULSNAR, classifier performance improved significantly (“Experiments and Results”) for SCAR and SNAR data, respectively.

An important contribution of this work is the speed of α estimation. Our experimentation showed that the KM1, KM2, and TIcE algorithms exhibited scalability issues. This observation aligns with the findings of Garg et al. (2021), who noted the underperformance of these techniques in high-dimensional scenarios and scalability issues with large datasets. While we evaluated PULSCAR/PULSNAR against these methods using moderately sized datasets, it is plausible that their inherent limitations with data size and high dimension contributed to inaccurate α estimates for some of our test datasets. Considering the scalability limitations and long execution time associated with KM1, KM2, TIcE, and DEDPUL (Table 2), these algorithms would be challenging to use for real-world applications involving millions of records and thousands of features. On the contrary, our preliminary work (Kumar et al., 2022) suggests that PULSCAR and PULSNAR are suitable for processing very large high-dimensional datasets ( n>1M, 150,000+ features). While in principle, other SCAR-based algorithms can be substituted for PULSCAR into our PULSNAR clustering approach, due to overestimation/underestimation issues (Figs. 3A, 4), scalability concerns, maintenance issues, and execution time constraints (Table 2), employing KM1, KM2, TIcE, and DEDPUL for estimating α in SNAR settings was not implemented in this work.

A limitation of our approach is that it relies on knowing whether the data are SCAR or SNAR to effectively select between PULSCAR or PULSNAR for α estimation. The issue is the “knee point detection” cluster determination approach relies on three consecutive points in the BIC curve to identify the knee (angle). As the first point cannot have a knee, this approach always returns more than one cluster. We tried other methods, such as minimum BIC, difference in BIC, or other clustering algorithms, but saw performance degradation in PULSNAR α estimates. When more than one clusters are used with PULSNAR on SCAR data, it can result in PULSNAR overestimating α as two near-identical positive types cannot be distinguished and get counted more than once. This α overestimation would adversely affect classification performance. In the future, we aim to enhance the PULSNAR algorithm so that it can be applied without knowing whether the data are SCAR or SNAR. Importantly, preliminary work indicates that PULSNAR α estimation is robust to overestimating the number of clusters in SNAR data: additional clusters beyond the known number of subtypes just slightly increase α estimates. The primary focus of our study was the estimation of α under the SCAR assumption failure. Preliminary findings on cluster overestimation, though informative, were excluded to streamline the narrative and emphasize the core findings.

The utility of α estimation might be illustrated with the problem of estimating the fraction of bots among social media accounts, an active area of ML research (Efthimion, Payne & Proferes, 2018; Heidari, James & Uzuner, 2021). If a trustworthy α upper bound estimate for the bot fraction among unlabeled accounts came to, say, 20%, business decisions could be made based on accounts comprising at least 80% real people. However, one might expect that positively detected bots are not SCAR, and thus, PULSNAR provides a means for estimating α despite that. However, with the PULSNAR clustering heuristic, we cannot guarantee an upper bound as we can in the SCAR case. Nevertheless, our experiments show that PULSNAR far outperformed SCAR-based methods with SNAR positives. In addition, we have found high utility with PULSNAR in identifying uncoded self-harm in electronic health records (Kumar et al., 2022). In that setting, PULSCAR underestimated α due to self-harm positives being SNAR as expected, but with PULSNAR, as confirmed with chart review, we had accurate α estimation as well as good calibration. Notably, that effort identified 14 and 15 self-harm clusters in two independent datasets respectively, demonstrating PULSNAR’s effectiveness in subclass clustering for accurate α estimates. Since the SCAR assumption frequently does not hold in real-world data, robust calibrated model estimations along with α estimates in SNAR settings using PULSNAR opens up new horizons in PU Learning.

Appendix

Algorithm for calibrating probabilities

Under the SCAR assumption, the relationship p(y=1|x)=p(s=1|x)p(s=1|y=1) holds, where p(s=1|y=1) is a constant (Lemma 1 from Elkan & Noto (2008)). Although this relationship provides a theoretical basis, it does not guarantee that the estimated probabilities, p(y=1|x), are well-calibrated, particularly in imbalanced datasets. To address this limitation, we propose a novel algorithm designed to effectively calibrate probabilities in both balanced and imbalanced datasets.

Algorithm 4 shows the complete pseudocode to calibrate the classifier-estimated class 1 probabilities of PU examples. Once α is known, we seek to transform the original class 1 probabilities so that their sum is equal to α|U| among the unlabeled or |P|+α|U| among positive and unlabeled, and that they are well-calibrated. Our approach is to probabilistically flip the assigned label of α|U| unlabeled examples to positive (from 0 to 1) in such a way as to match the PDF of labeled positives across 100 equispaced bins over [0…1], then fit a logistic or isotonic regression model on those labels versus the classifier-estimated class 1 probabilities to generate the transformed probabilities. To determine the number of unlabeled examples that need to be flipped in each bin, we compute the normalized histogram density, D_hist, for the labeled positives with 100 bins and then multiply α|U| with D_hist.

Algorithm 4 Calibrate probabilities (calibrate_probabilities).

Input: probs, s = sp∪su, n_bins, calibration_method, calibration_data, α	
{‘probs’ indicates the classifier-estimated class 1 probabilities for all examples; ‘s’ indicates assigned labels (labeled positives: s=1, unlabeled: s=0); n_bins is number of bins}	
Output: calibrated_probs	
  1: pu← probs[s == 0] {s==0 indicates unlabeled indices}	
  2: pp← probs[s == 1] {s==1 indicates labeled positive indices}	
  3: Dhist← histogram(pp, n_bins, density=True)	
  4: unlab_pos_count_in_bin ←α|pu|Dhist	
  5: binsu← split unlabeled examples into n_bins using pu	
  6: for k ← [n_bins … 1] do	
  7:    n1← unlab_pos_count_in_bin[k]	
  8:    n2←binsu[k]	
  9:   if n1>n2 then	
 10:    su^← flip labels (su) of n2 examples from 0 to 1 in bin k	
 11:   unlab_pos_count_in_bin[k-1] ← unlab_pos_count_in_bin[k-1] + ( n1−n2)	
 12:  else	
 13:    su^← flip labels (su) of random n1 examples from 0 to 1 in bin k	
 14:  end if	
 15: end for	
 16: if calibration_data == ‘PU’ then	
 17:  p, s^←pp∪pu, sp∪su^	
 18: else if calibration_data == ‘U’ then	
 19:  p, s^←pu, su^	
 20: end if	
 21: if calibration_method is ‘sigmoid’ then	
 22:   p^← LogisticRegression(p, s^)	
 23: else if calibration_method is ‘isotonic’ then	
 24:   p^← IsotonicRegression(p, s^)	
 25: end if	
 26: return p^	

The unlabeled examples are also divided into 100 bins based on their estimated probabilities. Starting from the bin with the highest probability (p = 1), we randomly select m examples and flip their labels from 0 to 1, where m is the number of unlabeled examples that need to be flipped in the bin. If the number of records ( n1) that need to be flipped in a bin is more than the number of records ( n2) present in the bin, the difference ( n1−n2) is added to the number of records to be flipped in the next bin, resulting in α|U| flips.

After flipping the labels of α|U| unlabeled examples from 0 to 1, we fit an isotonic or sigmoid regression model on the classifier-estimated class 1 probabilities with the updated labels to obtain calibrated probabilities.

The above calibration approach applies to both SCAR and SNAR data. For the SNAR data, the PULSNAR algorithm divides labeled positive examples into c clusters and estimates the α for each cluster. For each cluster, the classifier-estimated class 1 probabilities of the examples (positives from the cluster and all unlabeled examples or only unlabeled examples) are calibrated using the estimated α for the cluster. Since, for each cluster, PULSNAR uses all unlabeled examples, each unlabeled example has c classifier-estimated/calibrated probabilities. The final classifier-estimated/calibrated probability of an unlabeled example is calculated using the following Eq. (6):

(6) p=1−(1−p1)(1−p2)…(1−pc)

where pc is the probability of an unlabeled example from cluster c. Since we do not know the subclass of an unlabeled example, this Eq. (6) calculates one minus the probability that it is in none of the subclasses, and constrains the probability to be ≤1.

Experiments and results

We used synthetic SCAR and SNAR datasets and the KDD Cup SCAR dataset to test our calibration algorithm.

SCAR datasets: After estimating the α using the PULSCAR algorithm, we applied Algorithm 4 to calibrate the classifier-estimated probabilities. To calculate the calibrated probabilities for both positive and unlabeled (PU) examples, we applied isotonic regression to the classifier-estimated class 1 probabilities of PU examples with labels of positives and updated labels of unlabeled (of which α|U| were flipped per Algorithm 4). We applied isotonic regression to the unlabeled’s estimated probabilities with their updated labels to calculate the calibrated probabilities only for the unlabeled.

SNAR datasets: Using the PULSNAR algorithm, the labeled positive examples were divided into c clusters. For each cluster, after estimating the α, Algorithm 4 was used to calibrate the classifier-estimated probabilities. To calculate the calibrated probabilities for positives from a cluster and all unlabeled examples, we applied isotonic regression to their classifier-estimated class 1 probabilities with labels of positives from the cluster and updated labels of unlabeled (of which αj|U| were flipped for cluster j=1…c, see Algorithm 4). We applied isotonic regression to the unlabeled’s estimated probabilities with their updated labels to calculate the calibrated probabilities only for the unlabeled. Thus, each unlabeled example had c calibrated probabilities. We computed the final calibrated probability for each unlabeled example using Eq. (6).

Figures 6–11 show the calibration curves generated using the unblinded labels and isotonically calibrated (red)/uncalibrated (blue) probabilities. When both positive and unlabeled examples were used to calculate calibrated probabilities, the calibration curve followed the y = x line (well-calibrated probabilities). When only unlabeled examples were used, the calibration curve for 1% did not follow the y = x line, presumably due to the ML model being biased toward negatives, given the small α. Also, the calibration curves for the SCAR data followed the y = x line more closely than the calibration curves for the SNAR data. It is due to the fact that the final probability of an unlabeled example in the SNAR data is computed using its c probabilities from c clusters. So, a poor probability estimate from even one cluster can influence the final probability of an unlabeled example. Importantly, the sum of the calibrated probabilities adds up to |P|+α|U| when both positive and unlabeled examples are calibrated, and to α|U| when only unlabeled examples are calibrated, despite some of the calibration curves appearing to be skewed above the target y=x.

Figure 6 PULSCAR: calibration curves for synthetic SCAR datasets (both positive and unlabeled examples).

Synthetic datasets were generated with different fractions of positives (1%, 5%, 10%, 20%, 30%, and 50%) among the unlabeled examples. class_sep = 0.3, number of attributes = 100, |P| = 5,000 and |U| = 50,000. Calibration curves were generated using both positive and unlabeled examples (Uncalibrated probabilities-blue, calibrated probabilities-red).

Figure 7 PULSCAR: calibration curves for synthetic SCAR datasets (only unlabeled examples).

Synthetic datasets were generated with different fractions of positives (1%, 5%, 10%, 20%, 30%, and 50%) among the unlabeled examples. class_sep = 0.3, number of attributes = 100, |P| = 5,000 and |U| = 50,000. Calibration curves were generated using only unlabeled examples (Uncalibrated probabilities-blue, calibrated probabilities-red).

Figure 8 PULSNAR: calibration curves for synthetic SNAR datasets (both positive and unlabeled examples).

Synthetic datasets were generated with different fractions of positives (1%, 5%, 10%, 20%, 30%, and 50%) among the unlabeled examples. class_sep = 0.3, number of attributes = 100, number of positive subclasses = 5, |P| = 20,000 (4,000 from each subclass) and |U| = 50,000. Calibration curves were generated using both positive and unlabeled examples (Uncalibrated probabilities-blue, calibrated probabilities-red).

Figure 9 PULSNAR: calibration curves for synthetic SNAR datasets (only unlabeled examples).

Synthetic datasets were generated with different fractions of positives (1%, 5%, 10%, 20%, 30%, and 50%) among the unlabeled examples. class_sep = 0.3, number of attributes = 100, number of positive subclasses = 5, |P| = 20,000 (4,000 from each subclass) and |U| = 50,000. Calibration curves were generated using only unlabeled examples (Uncalibrated probabilities-blue, calibrated probabilities-red).

Figure 10 PULSCAR: calibration curves for SCAR KDD Cup 2004 particle physics dataset (both positive and unlabeled examples).

Unlabeled sets contained 1%, 5%, 10%, 20%, 30%, and 40% positive examples. Calibration curves were generated using both positive and unlabeled examples (Uncalibrated probabilities-blue, calibrated probabilities-red).

Figure 11 PULSCAR: calibration curves for SCAR KDD Cup 2004 particle physics dataset (only unlabeled examples).

Unlabeled sets contained 1%, 5%, 10%, 20%, 30%, and 40% positive examples. Calibration curves were generated using only unlabeled examples (Uncalibrated probabilities-blue, calibrated probabilities-red).

Improving classification performance with pulscar and pulsnar

Algorithm 5 shows the complete pseudocode to improve classification performance with PULSCAR and PULSNAR. The algorithm returns the following six classification metrics: Accuracy, AUC-ROC, Brier score (BS), F1, Matthew’s correlation coefficient (MCC), and Average precision score (APS). The approach to enhancing the classification performance is as follows:

Algorithm 5 Calculate classification metrics.

Input: X = Xp∪Xu, s = sp∪su, y_true, calibration_method, n_bins, probs, α	
{‘probs’ indicates the classifier-estimated class 1 probabilities for all examples; ‘s’ indicates assigned labels (labeled positives: s=1, unlabeled: s=0); n_bins is number of bins}	
Output: classification_metrics (accuracy, roc auc, brier score, f1, Matthew’s correlation coefficient, average precision)	
 1: p^← calibrate_probabilities(probs, s, n_bins, calibration_method, ‘U’, α)	
 2: sort p^ in descending order	
 3: su^← flip labels (su) of top α|U| unlabeled examples with highest p^ from 0 to 1	
 4: s^←sp∪su^	
 5: p ←A(X, s^) { A(X, s^) returns estimated class 1 probabilities for all examples}	
 6: return accuracy(p, y_true), auc(p, y_true), bs(p, y_true), f1(p, y_true), mcc(p, y_true), aps(p, y_true)	

Using PULSCAR: After estimating the α, the classifier-estimated (classifier trained on labeled positive and unlabeled data) class 1 probabilities of only unlabeled examples are calibrated using Algorithm 4. The calibrated probabilities of the unlabeled examples are sorted in descending order, and the labels of top α|U| unlabeled examples with the highest calibrated probabilities are flipped from 0 to 1 (probable positives). We then train and test a second ML classifier (XGBoost) with five-fold CV using the labeled positives and probable positives as class 1, and the remaining unlabeled examples as class 0. The classification performance metrics are calculated using the second classifier-estimated class 1 probabilities and the true labels of the examples.

Using PULSNAR: The PULSNAR algorithm divides the labeled positive examples into c clusters. For each cluster, after estimating αj for j in 1…c, the classifier-estimated (classifier trained on labeled positive and unlabeled data) class 1 probabilities of only unlabeled examples are calibrated using Algorithm 4. Since each unlabeled example has c calibrated probabilities, we compute the final calibrated probability for each unlabeled example using the Eq. (6). The final α is calculated by summing the αj values over the c clusters. The final calibrated probabilities of the unlabeled examples are sorted in descending order, and the labels of top α|U| unlabeled examples with the highest calibrated probabilities are flipped from 0 to 1 (probable positives). We then train and test a second ML classifier (XGBoost) with five-fold CV using the labeled positives and probable positives as class 1, and the remaining unlabeled examples as class 0. The classification performance metrics are calculated using the second classifier-estimated class 1 probabilities and the true labels of the examples.

Experiments and results

We applied Algorithm 5 to synthetic SCAR and SNAR datasets to get the performance metrics for the XGBoost model with PULSCAR and PULSNAR, respectively. The classification performance metrics were also calculated without applying the PULSCAR or PULSNAR algorithm, in order to determine the improvement in the classification performance of the model. The experiment was repeated 40 times by selecting different train and test sets using 40 random seeds to compute the 95% confidence interval (CI) for the metrics.

Figures 12 and 13 show the classification performance of the XGBoost model with/without the PULSCAR or PULSNAR algorithm on synthetic SCAR and SNAR data, respectively. The classification performance using PULSCAR or PULSNAR increased significantly over XGBoost alone. As the proportion of positives among the unlabeled examples increased, the performance of the model without PULSCAR or PULSNAR (blue) worsened significantly more than when using PULSCAR or PULSNAR.

Figure 12 Classification performance of XGBoost model on synthetic SCAR datasets with and without the PULSCAR algorithm.

Synthetic datasets were generated with different fractions of positives (1%, 5%, 10%, 20%, 30%, 40%, and 50%) among the unlabeled examples. class_sep = 0.3, number of attributes = 100, |P| = 5,000 and |U| = 50,000. “without PULSCAR” (brown): XGBoost model was trained and tested with five-fold CV on the given data; the classification metrics were calculated using the model estimations and true labels. “with PULSCAR” (blue): PULSCAR algorithm was used to find the proportion of positives among unlabeled examples ( α); using α, probable positives were identified; XGBoost model was trained and tested with five-fold CV on labeled positives, probable positives, and the remaining unlabeled examples; classification metrics were calculated using the model estimations and true labels. The error bars represent 95% CIs for the performance metrics.

Figure 13 Classification performance of XGBoost model on synthetic SNAR datasets with and without the PULSNAR algorithm.

Synthetic datasets were generated with different fractions of positives (1%, 5%, 10%, 20%, 30%, 40%, and 50%) among the unlabeled examples. class_sep = 0.3, number of attributes = 100, number of positive subclasses = 5, |P| = 20,000 (4,000 from each subclass) and |U| = 50,000. “without PULSNAR” (brown): XGBoost model was trained and tested with five-fold CV on the given data; the classification metrics were calculated using the model estimations and true labels. “with PULSNAR” (blue): PULSNAR algorithm was used to find the proportion of positives among unlabeled examples ( α); using α, probable positives were identified; XGBoost model was trained and tested with five-fold CV on labeled positives, probable positives, and the remaining unlabeled examples; classification metrics were calculated using the model estimations and true labels. The error bars represent 95% CIs for the performance metrics.

Dedpul vs. pulsnar: alpha estimation

Public implementations of the PU learning methods KM1, KM2, and TIcE were not scalable; they either failed to execute or would have taken weeks to run the multiple iterations required to obtain confidence estimates for large datasets. We thus could not compare our method with KM1, KM2, and TIcE on large datasets and used only DEDPUL for comparison. Importantly, it was previously demonstrated that the DEDPUL method outperformed these three methods on several ML benchmark and synthetic datasets (Ivanov, 2020).

We compared our algorithm with DEDPUL on synthetic SNAR datasets with different fractions (1%, 5%, 10%, 20%, 30%, 40%, and 50%) of positives among unlabeled examples. In our experiments, we observed that class imbalance (ratio of majority class to minority class) could affect the α estimates. So, we used four different sample sizes: 1) positive: 5,000 and unlabeled: 5,000; 2) positive: 5,000 and unlabeled: 25,000; 3) positive: 5,000 and unlabeled: 50,000; 4) positive: 5,000 and unlabeled: 100,000. For each sample size and fraction, we generated 20 datasets using sklearn’s make_classification() method with random seeds 0–19 to compute 95% CI. We used class_sep = 0.3 for each dataset to create difficult classification problems. All datasets were generated with 100 attributes and six labels (0–5), defining ‘0’ as negative and 1–5 as positive subclasses. The positive set contained 1,000 examples from each positive subclass in all datasets. The unlabeled set comprised k% positive examples with labels (1–5) flipped to 0 and (100-k)% negative examples. The unlabeled positives were markedly SNAR, with the five subclasses comprising 1/31, 2/31, 4/31, 8/31, and 16/31 of the unlabeled positives.

Figure 14 shows the α estimates by DEDPUL and PULSNAR on synthetic SNAR data. For smaller true fractions (1%, 5%, 10%), DEDPUL returned close α estimates, but for larger fractions (20%, 30%, 40%, and 50%), it underestimated α. Also, as the class imbalance increased, the performance of DEDPUL dropped, especially for larger true fractions. The estimated α by the PULSNAR method was close to the true α for all fractions and sample sizes.

Figure 14 (A–D) PULSNAR and DEDPUL evaluated on synthetic SNAR datasets.

The bar represents the mean value of the estimated α, with 95% CI for estimated α. The best estimators are close to the black bars, representing the true α. Bars larger than the black bars represent overestimation, while bars smaller than the black bars represent underestimation.

Dedpul vs. pulsnar: classification performance

The main focus of this study was on comparing α estimates provided by different PU methods. Hence, we did not extensively compare the classification performance of our methods with other PU methods. Moreover, methods such as KM1, KM2, and TIcE do not return estimated probabilities, which hinders the comparison of classification performance with these methods. Also, the classification performance of PULSCAR and DEDPUL on SCAR datasets was comparable. Consequently, we compared the classification performance of PULSNAR and DEDPUL using two SNAR datasets: smartphone and dry bean. Future studies would focus on a more comprehensive comparison of classification performance and exploring further enhancements in this regard.

For DEDPUL, leveraging its estimated α, we flipped the labels of α|U| unlabeled examples with the highest estimated probabilities from 0 to 1. Subsequently, we employed the XGBoost model with five-fold CV, treating true positives and probable positives as positive (class 1) examples, and probable negatives as negative (class 0) examples to generate estimated class 1 probabilities for all examples in the datasets. The classification performance metrics were then computed using these XGBoost estimated probabilities and the true labels of the examples.

For PULSNAR, we employed the method discussed in “Improving Classification Performance with PULSCAR and PULSNAR” to calculate the classification performance metrics.

Figures 15 and 16 show the classification performance of the XGBoost models with/without identifying probable positives using DEDPUL and PULSNAR. As the true fraction of positives among the unlabeled set increased for both datasets, the α estimates by DEDPUL were not close to true fractions, leading to a decrease in its classification performance. No significant drop in the classification performance of PULSNAR was observed, as the α estimates by PULSNAR were close to true answers.

Figure 15 Classification performance of XGBoost model on dry bean SNAR dataset with/without DEDPUL and PULSNAR.

“without PU” (brown): performance of XGBoost model without applying DEDPUL or PULSNAR to identify probable positives. “with DEDPUL” (golden): performance of XGBoost model after identifying probable positives using DEDPUL. “with PULSNAR” (blue): performance of XGBoost model after identifying probable positives using PULSNAR. The error bar represents a 95% confidence interval of the metric.

Figure 16 Classification performance of XGBoost model on smartphone SNAR dataset with/without DEDPUL and PULSNAR.

“without PU” (brown): performance of XGBoost model without applying DEDPUL or PULSNAR to identify probable positives. “with DEDPUL” (golden): performance of XGBoost model after identifying probable positives using DEDPUL. “with PULSNAR” (blue): performance of XGBoost model after identifying probable positives using PULSNAR. The error bar represents a 95% confidence interval of the metric.

Additional Information and Declarations

Competing Interests

Author Contributions

Data Availability

1 A significant portion of the text and experimental results presented in this manuscript was previously published as part of a preprint (https://arxiv.org/abs/2303.08269; Kumar & Lambert, 2023).

The authors declare that they have no competing interests.

Praveen Kumar conceived and designed the experiments, performed the experiments, analyzed the data, performed the computation work, prepared figures and/or tables, authored or reviewed drafts of the article, and approved the final draft.

Christophe G. Lambert conceived and designed the experiments, performed the experiments, analyzed the data, performed the computation work, prepared figures and/or tables, authored or reviewed drafts of the article, and approved the final draft.

The following information was supplied regarding data availability:

The full source code for our algorithms (and the code to generate all of the simulated datasets) is available at GitHub and Zenodo:

- https://github.com/unmtransinfo/PULSNAR.

- Praveen Kumar, & Christophe Lambert. (2024). unmtransinfo/PULSNAR: PULSNAR 0.0.1 (PULSCAR). Zenodo. https://doi.org/10.5281/zenodo.13126647.

The code that generates the SCAR simulated data is available at GitHub: https://github.com/unmtransinfo/PULSNAR/blob/main/examples/pulscar_syn_alpha_estimation_1.py.

This code generates the SNAR simulated data is available at GitHub: https://github.com/unmtransinfo/PULSNAR/blob/main/examples/pulscar_syn_alpha_estimation_2.py.

The third-party SCAR data and SNAR data are available at GitHub: https://github.com/unmtransinfo/PULSNAR/tree/main/examples/UCIdata

The third party datasets are available at:

- KDDCup 2004 repository: https://kdd.org/kdd-cup/view/kdd-cup-2004

- UCI Machine learning repository: https://archive.ics.uci.edu/datasets.

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
