# Peer review of "Positive Unlabeled Learning Selected Not At Random (PULSNAR): class proportion estimation without the selected completely at random assumption"

_PeerJ Computer Science, doi:10.7717/peerj-cs.2451_

## Round 0.1 · original submission · Major Revisions

All three reviewers agree that your paper is interesting but identified some issues that need to be addressed before it can be published. Therefore, I recommend the paper undergo a "major revision."

Please take into consideration all the suggestions and concerns raised by the reviewers. In particular, I notice that most of them agree there are several aspects in the paper that need clarification, including some fundamental definitions and assumptions.

We look forward to receiving your revised manuscript.

Reviewer 1 ·

Basic reporting

The language of the paper could be improved in some parts. For example,
- line 64: evaluate instead of evaluating
- lines 47 and 65: medium-sized instead of modest
- line 114: a method instead of method

Some statements in the paper are confusing:
- line 148: "The goal is to obtain machine learning-predicted probabilities for all instances ..." Probabilities of what event?
- line 153: "... any ML algorithm, A(x), that generates [0...1] probabilities for the data ..." Probabilities of what event?
- lines 202 and 203: "Let `pr' be the ML-predicted probabilities of PU examples ..." Probabilities of what event?
- line 609: "we fit an isotonic or sigmoid regression model on the ML-predicted class 1 probabilities ..." Where do these probabilities come from?

The paper is not necessarily self-contained. Only after having read Elkan & Noto (2008) and Ivanov (2020) I began to understand what it is about.

Some terms and notions are unexplained or rather vague:
- lines 144 and 145: "Unlabeled positive instances have analogous counterparts (examples with similar features) in the labeled positive set." What does
"analogous counterpart" exactly mean?
- Figure 1: "case", "estimated control", "true control" are not explained.
- line 592: The term p(y = 1|(s = 1|x)) is not properly defined, hence it is unclear why it should be equal to 1.

Experimental design

Equation (3): The rationale for choosing the objective function f(\alpha) is not well explained. According to Algorithm 1,
\alpha is determined such that the value of the first derivative f'(\alpha) "changes maximally". Does this mean that the second derivative f''(\alpha) is maximised?
Line 159 and 160 state: "PULSCAR estimates \alpha by finding the value \alpha where the following objective function [i.e. f(\alpha)] maximally
changes". This would imply that the first derivative f'(\alpha) is maximised. Please clarify. In any case it is not very clear why this approach should be more efficient than and
deliver the same solution as calculating \alpha = min_x f_u(x) / f_p(x).

Algorithm 1: What kind of predicted probabilities does A(X,y) calculate? Why are negative instances y==0 known? That seems to be in
contradiction to the general assumptions of the paper.

Line 263: Why is there a need to learn another classifier when calibrated posterior probabilities of the positive class, i.e. a Bayes classifier has been found?

Appendix A: Why is a proof needed if by definition SNAR means "not SCAR", i.e. p(s = 1|x, y = 1) is not constant?

Appendix B: In the SCAR case, according to Lemma 1 of Elkan & Noto (2008) the posterior probabilities of the positive class are just a constant multiple of the
"ML-predicted probabilities of positive and unlabeled examples". Hence why is there a need for such a complicated algorithm as described in Appendix B?

Line 597: "flip ... labels of unlabeled to positive (from 0 to 1)". Sounds strange. If an instance is unlabeled its label can be assumed to be 1 but
it cannot be flipped as there is none.

Equation (6): Why not just p = sum_i p_i? Because then p > 1 might happen? But for (6) to make sense it might be necessary to assume that the events
"in subclass i" are conditionally independent given the covariates -- which is not true as they depend on each other by the constraint sum_i p_i <= 1.

Validity of the findings

Figure 3: Algorithm TICE appears to be spectacularly wrong -- which is somewhat implausible. Was it correctly deployed?

Figure 9: The charts for the calibrated probs suggest that the PULSNAR calibration procedure is systematically biased. Would you recommend the approach
nonetheless?

Additional comments

Disclosure: I read the reviewer comments and the authors' responses on https://openreview.net/forum?id=L4lHDXzcDu

Regarding PULSNAR, I feel that the previous reviewers' questions were satisfactorily answered, with corresponding amendments in the paper.

I'm still sceptical with regard to the choice of the objective function (eq. (3)) for PULSCAR although it seems to work very well in practice.

Regarding PULSNAR I'm sceptical mainly because of Figure 9. Perhaps `divide and conquer', dealing with one cluster only at a time, is not optimal and results
in poor calibration. Do you see any chance for jointly estimating the alphas for the different clusters?

·

Basic reporting

This paper presents machine learning algorithms to estimate the proportion of positive examples in an unlabeled dataset, given that we are provided with a labeled set of only positive examples and a set of unlabeled data (Positive and Unlabeled (PU) learning). The paper is general is interesting, as are the proposed methods and its applications. Even though the paper is well structured and the language clear, I have serious concerns about how the methods are explained, in particular the inner workings of the PULSCAR method. Here are some general comments about the paper:

• Lines 50 to 54. Authors state that the SCAR assumption can be violated in real problems. I think some other examples can be added (apart from the spam one) to illustrate it, given that this is one of the main points of the paper.

• Lines 132 to 140. In this paragraph the authors set some conditions that would make the SCAR assumption fail, giving some nice examples. I think this part can be further formalized. My intuition is that the key ingredient here is P(X|y), that should remain constant for the positive class between the labeled and unlabeled datasets. Once you have subclasses under one class and they do not vary uniformly in both datasets, P(X|y) will change.

• Lines 147 to 149. As far as I understand, the proposed algorithms work using the predictions of a probabilistic classifier. The text should further clarify how this classifier is trained. I think that you are training the classifier using positives (from the labeled set) and unlabeled (which contains a mix of positives and negatives). This classifier would be trained then to predict if an example is labeled or unlabeled. What would happen if positives are the prevalent class in the unlabeled set? How this would affect the probabilities returned by this classifier? Would this pose a problem to your methods?

• When explaining PULSCAR algorithms, there are some elements that make the explanation difficult to follow. I would try to summarize the main ones:
o PDFs are represented with f(x) in the text, but then they are represented with the letter D in other parts of the text (algorithm). In Figure 1, which is key to understand the algorithm, these letters do not appear any more, but some legend which is difficult to understand. I would be in favor for using the D notation everywhere when you refer to the PDFs of the posterior probabilities returned by a probabilistic classifier. Maybe it would be nice to add a section with the problem definition and notation used in the paper.
o A key part of the algorithm is that you look for the smallest value of alfa where D_u-alfa*D_p is always positive in [0,1], to have an upper bound of alfa. This is understood by the readers when you get to Figure 1 but it should be clarified before in the text.
o Algorithm 1 should revised. In Algorithm 1, authors state that the input of the algorithm is X_p, X_u, y_p and y_u. As far as I understand that the input of the algorithm are the examples of both labeled and unlabeled set, plus the y_hat (predicted probability) given by the probabilistic classifier. First, I would not use the symbol y, as it seems it is the real class (which is not known for the unlabeled). Secondly, in line 1 of the algorithm you compute the predicted probabilities, so it does not make sense to me to receive y_p and y_u (I think you are computing that inside the algorithm). Even more confusing is that the classifier is taking as an input both X and y. Also label 0 and label 1 are confusing in this context (an in other parts of the text). In my case, I had to go to the actual code to try to understand how this actually works.

• Lines 237 to 259. The clustering idea in SNAR is nice, given that you are able to find some subclasses that make the learning assumptions of SCAR hold (basically that P(X|y) is constant for positive data between labeled and unlabeled datasets. In my opinion, the critical part here is doing a good job clustering, which might be challenging in many situations. I think I would be interesting to see in the experiments how well the clustering algorithms where able to identify the subclasses in the data, and how this relates to performance of PULSNAR.

• About the results, I have some comments:
o What is the problem with TICE method that is unable to work in the synthetic data. Have you analyzed this?
o It looks to me that for SCAR synthetic data, DEDPUL method is working better in almost all cases (this also happens for instance in KDD cup 2004 data). It would be nice to include some results table which sometimes results can be better interpreted that using a single chart.


Other minor issues:
• The paper uses the word ML algorithm many times. I would change it by probabilistic classifier as it is clearer.
• Line 173, where it says “class 1 probabilities of positives and unlabeled”, what is the meaning of “class 1”?

Experimental design

Nothing to add

Validity of the findings

Nothing to add

Additional comments

Nothing to add

Reviewer 3 ·

Basic reporting

The paper introduces two methods for Positive-Unlabeled (PU) classification: PULSCAR, tailored for the Selected Completely At Random (SCAR) assumption, and PULSNAR, developed for the Selected Not At Random (SNAR) assumption. Both methods demonstrate promising performance.

The paper has some issues in my opinion:

1) The definition of PU data assumptions in Page 3 can be improved.

Assumption 1: “Positive examples have correct labels, meaning no negative example is marked as positive.”

This defines the PU classification problem itself, but your sentence seems to imply that the training set is noise-free. Noise occurs in real problem, and examples are sometimes incorrectly labeled. I believe this assumption is unnecessary.

Assumption 2: “Unlabeled positive instances have analogous counterparts (examples with similar features) in the labeled positive set. If the second assumption does not hold, our approaches may underestimate \alpha.”

In the literature, this is typically referred to as “support”, meaning that unlabeled positive instances have “support” in the labeled positive set, in symbols: if f_u+(x)!=0, this implies that f_p(x)!=0 as well. In fact, looking Equation 2, the actual assumption seems even stronger: p(x|y=1,s=1)=p(x|y=1,s=0). Otherwise, it is unclear how f_u(x) can be defined as a mixture of f_p(x) proportional to \alpha. However, the SCAR assumption precisely ensures this, because if SCAR assumption holds, the set of labeled examples is an i.i.d. sample from the positive distribution, f_u+(x) = f_p(x).

Considering the emphasis you place on this assumption, I have one question:

- You mention that your approaches may underestimate \alpha. Do they never overestimate \alpha? Why? According to the reported results, such as those in Figure 3, PULSCAR sometimes overestimates alpha.

2) You can also state the assumption made by PULSNAR method more clearly. Essentially, the assumption is that the data is SNAR, but the positive class consists of several subclasses, each obeying the SCAR assumption. Right?

3) The justification for excluding those methods in the literature designed under the SNAR assumption is inadequate. Although such methods do not estimate \alpha or calibrate the probabilities, you can still compare their classification performance, which is the main goal of PU classification tasks.

4) Did you consider analyzing the classification performance of PULSNAR on SCAR data? It would be interesting.

5) Experiment in Figure 5: the comparison is not fair. All methods, except PULSNAR, are devised for SCAR assumption. You should include other methods, see comment 3 above.

Experimental design

no comment

Validity of the findings

no comment

Additional comments

- Legends in Figure 1 should be explained. They are difficult to understand, e.g., “case”.

- The paper has too many subsections, especially in the Experimental Methods and Results sections.

---

## Round 0.2 · Minor Revisions

Two of the three reviewers have already recommended the paper for publication. However, Reviewer #1 still raises some concerns and recommends another round of major revisions. After carefully reviewing Reviewer #1's feedback, I believe the paper is ready for publication once some minor issues are addressed. Therefore, I will be recommending a final round of minor revisions, which I will personally review to avoid further delays in the publication process.

Please also take this opportunity to improve the mathematical rigor of the presentation, as one of the reviewers has critiqued this aspect.

Reviewer 1 ·

Basic reporting

Comments on revised version

Lines 161 and 162: "In the SNAR case where ..."
Make it clear that this is not a conclusion but an assumption on a special case of SNAR. SNAR cases with other structures might be conceivable.

Lines 166 to 171:
In this paragraph, you could make it clear once and for all that you are dealing with two different classification problems: A (first run) problem to distinguish labeled (s=1) vs. unlabeled (s=0) instances. Then the second (main) problem to distinguish positive (y=1) vs. negative (y=0) instances.
Perhaps you should stop talking about a positive class when referring to the labeled vs. unlabeled classification as doing so is likely to create confusion.

Line 173: "Given any supervised probabilistic classifier, A (x, y),"
No need to introduce A(x,y) already here. Only creates confusion.

Line 179: "i \in [0 ... 1]"
If i refers to elements of a sample than it having range [0,1] is strange.

Line 182: "Since Du and Dp are vectors"
No, Du and Dp are densities (i.e. functions).

Line 194: "Figure 1D"
Does Fig 1D really show f(alpha) as defined by (3)? If so, the curve should be flat for alpha >= 0.1 and be generally decreasing for alpha < 0.1.
Accordingly, one would rather expect your approach to underestimate the alpha_max than to overestimate it.

Algorithm 1: "A (X, s) returns estimated probability of being positive (class 1) for all examples"
Should be "probability of being labeled (class s=1)".

Algorithm 1: "absolute difference"
Do you mean to say "absolute value of the first derivative"? Hence you are looking for an inflection point of f(alpha)?

Appendix A
Please think about inserting here the rationale for not using the "constant multiplier" approach found by Elkan & Noto (which you provided in the response
to my comments on the previous version).

Experimental design

No further comments.

Validity of the findings

No further comments.

Additional comments

No further comments.

·

Basic reporting

I would like to thank the authors for addressing my concerns in their revised manuscript. After reviewing the changes, I am satisfied with the improvements and clarifications made in the responses to my comments. Below are my comments on the revisions:

Examples for SCAR Assumption (Lines 50-54): Adding real-world examples (patient health data, fraud detection, and social media bot detection) helps solidify the authors' argument that the SCAR assumption can be violated in practical scenarios. I think these additions improve the paper introduction and better contextualize the importance of addressing SCAR violations.

Formalization of SCAR conditional probabilities (Lines 132-140): In my opinion, the formalization of the failure of the SCAR assumption using conditional probabilities is a nice addition. This mathematical clarity helps the reader understand when and how the SCAR assumption breaks down.

Probabilistic Classifier and Imbalance (Lines 147-149): I appreciate the clarification on how the probabilistic classifier is trained and the reassurance that the method remains robust, even when positives dominate the unlabeled set.

Notational Consistency: The authors have addressed the notational inconsistencies I previously highlighted, and the revised manuscript is much clearer in this regard. The consistent use of the notation "D" for the PDFs and the improvements in Figure 1 make the explanation of the algorithm more accessible.

Algorithm 1 Revisions: The changes to Algorithm 1, including the replacement of "y" with "s" to avoid confusion and the clarification of how the classifier is used, have significantly improved the presentation.

Clustering in PULSNAR: The additional information regarding the clustering mechanism in PULSNAR and its performance in identifying subclasses is appreciated. The authors' explanation of how the number of clusters usually corresponds to the true subclasses in their experiments provides further confidence in the robustness of the method.

Overall, the revisions have addressed all of my concerns, and I think the paper is now much clearer. I believe the work is of high quality and would make a good contribution to the field of Positive and Unlabeled (PU) learning. I recommend acceptance of the paper in its current form.

Experimental design

See basic reporting

Validity of the findings

See basic reporting

Reviewer 3 ·

Basic reporting

I believe the article has improved after the first review, especially the part explaining the method: it is now clearer, some inconsistencies in the notation have been reduced, and the explanation of the figures has been improved. In my opinion, the experimental results are still not as convincing as they could be on the SNAR datasets, but despite this, the article is interesting enough to be published.

Minor comments:

Lines 176 and 180: Instead of "property 2," I would use "equation (2)," or at least put (2) instead of just 2 to make it clearer that it refers to equation 2.

Experimental design

See Basic report

Validity of the findings

See Basic report

Additional comments

See Basic report

---

## Round 0.3 · accepted · Accept

After reviewing the modifications and the response to reviewers myself, I believe the paper is ready for publication.
Congratulations!